# Biodegradable-Glass-Fiber Reinforced Hydrogel Composite with Enhanced Mechanical Performance and Cell Proliferation for Potential Cartilage Repair

**DOI:** 10.3390/ijms23158717

**Published:** 2022-08-05

**Authors:** Chenkai Zhu, Changyong Huang, Wuxiang Zhang, Xilun Ding, Yang Yang

**Affiliations:** 1School of Mechanical Engineering and Automation, Beihang University, Beijing 100191, China; 2Ningbo Institute of Technology, Beihang University, Ningbo 315832, China; 3School of Environmental and Biological Engineering, Nanjing University of Science & Technology, Nanjing 210094, China

**Keywords:** biodegradable glass fiber, poly(vinyl alcohol) hydrogel, composites, mechanical performance, degradation behavior

## Abstract

Polyvinyl alcohol (PVA) hydrogels are promising implants due to the similarity of their low-friction behavior to that of cartilage tissue, and also due to their non-cytotoxicity. However, their poor mechanical resistance and insufficient durability restricts their application in this area. With the development of biodegradable glass fibers (BGF), which show desirable mechanical performance and bioactivity for orthopedic engineering, we designed a novel PVA hydrogel composite reinforced with biodegradable glass fibers, intended for use in artificial cartilage repair with its excellent cytocompatibility and long-term mechanical stability. Using structure characterization and thermal properties analysis, we found hydrogen bonding occurred among PVA molecular networks as well as in the PVA–BGF interface, which explained the increase in crystallinity and glass transition temperature, and was the reason for the improved mechanical performance and better anti-fatigue behavior of the composites in comparison with PVA. The compressive strength and modulus for the PBGF-15 composite reached 3.05 and 3.97 MPa, respectively, equaling the mechanical properties of human articular cartilage. Moreover, the increase in BGF content was found to support the proliferation of chondrocytes in vitro, whilst the PVA hydrogel matrix was able to control the ion concentration by adjusting the ions released from the BGF. Therefore, this novel biodegradable-glass-fiber-reinforced hydrogel composite possesses excellent properties for cartilage repair with potential in medical application.

## 1. Introduction

The articular cartilage is a viscoelastic connective tissue, considered as one of the most efficient aqueous lubrication systems for load transmission and the articulation of joints [1]. However, due to the relatively avascular nature of the tissue, cells find it difficult to migrate within and repair the site, resulting in a lack of regenerative factors at the injury site in the cartilage and difficulty in healing once injury happens [2].

The biomaterials used to replace cartilage require sustained loading capability and comparable mechanical and biological compatibility. Compared with typical biomaterials for artificial cartilage, such as poly(ether-ether-ketone) (PEEK) and polyurethane (PU), poly(vinyl alcohol) (PVA) hydrogels exhibit desirable cytocompatibility with low friction coefficients, and have been recognized as candidates for cartilage regeneration [3,4], whereas their relatively low stiffness and strength render it unsuitable for use under the load-bearing conditions of joints [5]. Therefore, the addition of stiff fibers into PVA hydrogel is considered as a potential solution to improve mechanical capability [6,7]. 

Currently, the fibers used for PVA hydrogel composites are non-degradable, such as ramie fibers [8] and polyethylene (PE) fibers [9,10], of which the biocompatibility and bioactivity are limited in terms of cartilage repairing and tissue regeneration. With the development of bioactive glass, biodegradable glass fiber (BGF) with controllable degradation rate and tunable mechanical performance has been introduced [11,12]. The ionic components released from BGF during degradation present impressive capabilities for the promotion of cartilage cell proliferation and differentiation [13]. 

However, the current studies on BGF-reinforced composites involve rigid composite types based on the degradable polymer matrix, such as polylactic acid (PLA) [14] and polycaprolactone (PCL) [15]. Except for the bioglass-PVA hydrogel composite reported by Lin et al. [16], the soft hydrogel matrix with BGF reinforcement has been seldom reported, and the mechanism of composite degradation, mechanical performance variation, and bioactivity of the material are still unknown. Furthermore, the PVA hydrogel is composed of a three-dimensional network of cross-linking polymers and moisture in the gap, and has the ability to control ion migration, indicating its potential effect on the degradation of BGF and controllable ion release behavior [17,18].

Inspired by this, the authors of the present study prepared biodegradable glass fiber (BGF) reinforced PVA hydrogel composites (PBGF) using the freezing/thawing technique in order to avoid any potential toxic additives. The mechanical performance, structure stability, and bioactivity of the composites were expected to be improved significantly with the addition of the stiffer BGF, potentially resulting in better bioactivity for cell proliferation and anti-fatigue properties and compressive behaviors for cartilage in the joints. Thus, analyses of the intercalation behavior of the BGF toward the PVA hydrogel, the composite’s network formation, the dynamic compressive performance, and the composite’s bioactivity were emphasized in the current study.

## 2. Results and Discussion

### 2.1. Physical Structure and Properties of Composites

#### 2.1.1. Composite Structure Analysis

In this study, the biodegradable glass fibers (BGF) were employed to reinforce PVA hydrogel. As shown in Figure 1A and Table 1, ATR-IR analyses were performed to investigate the absorption-band spectra of biodegradable glass fiber (BGF), pure PVA, and composites reinforced by BGF with fiber mass fraction of 5% (PBGF-5), 10% (PBGF-10) and 15% (PBGF-15). A typical absorption peak (1143 cm^−1^) corresponding to the C-O-C stretching vibration of PVA [19,20] was observed, and peaks at 1750 cm^−1^, 2920 cm^−1^ and 3100 cm^−1^ to 3600 cm^−1^ were attributed to the stretching vibration of the C=O, C-H and O-H intermolecular bonds, respectively [21,22]. For BGF, the stretching vibration of the P-O-P bond in the glass structure was observed on a broad band from 750 cm^−1^ to 880 cm^−1^ and presented significantly on the spectra of the PBGF composites [23]. On the other hand, the vibration mode of the O-H bond gradually enhanced from PBGF-5 to PBGF-15 between 1300 cm^−1^ to 1380 cm^−1^, indicating the hydrogen bond that was formed between the BGF and the PVA hydrogel matrix [24,25]. 

The swelling capacity is critical to practical use in tissue engineering [26]. As shown in Figure 1B, all composites exhibited rapid swelling kinetics within the first 60 min, and then slowly increased over the 1440 min to reach to the equilibrium. As such, the addition of BGF could produce a lower swelling ratio, which could be attributed to BGF acting as a physical crosslinker leading to the generation of crosslink points in the polymeric network [27]. Thus, the network crosslinking density was improved with a more condensed structure, resulting in a lower swelling ratio. 

#### 2.1.2. Thermal Properties Analysis

In order to understand the variation in thermal properties introduced to composites by biodegradable glass fibers, thermogravimetric analysis (TGA) and differential scanning calorimetry (DSC) were performed. As shown in Figure 1C, composites with different fiber fractions showed different pyrolysis curves of TGA with two significant variation steps. The composites were observed to lose weight above 50 °C due to the loss of water in the hydrogel, and a steady loss was observed for temperatures above 177 °C. The second step for mass loss started from 270 °C, and was caused by pyrolysis of the PVA polymer. The final residual mass percentages of samples were 0.57%, 5.72%, 10.92% and 15.79% for the pure PVA hydrogel and the PBGF-5, PBGF-10 and PBGF-15 composites, respectively. These results are confirmed by Lin et al. [16], who reported that the addition of inorganic bioactive glass could reduce the pyrolysis behavior of composites, as inorganic components of bioactive glass did not undergo pyrolysis at high temperature.

In order to analyze the crystallinity, glass transition temperatures (T_g_), and melting points (T_m_) of composites, the DSC curves of PVA hydrogel and PBGF composites were reported in Figure 1D. The pure PVA hydrogel showed a relatively large and smooth endothermic curve, with T_g_ at 55.25 °C and T_m_ at 219.23 °C (Figure 1E). An increased trend in T_g_ in the composites was observed from 56.93 to 59.73 °C, suggesting that more crosslink formations in hydrogel networks was able to restrict motion in the hydrogel composites. The percentage of crystallinity (Figure 1F) for the PVA hydrogel composites calculated from DSC was revealed to increase slightly with the addition of BGF into the hydrogel matrix, where the pure PVA hydrogel exhibited the lowest percentage of crystallinity (2.85%), and the highest crystallinity was found in the PBGF-15 composite (7.48%), indicating significant effects on composite structure with the addition of BGF.

#### 2.1.3. The Mechanism of Composite Formation

As shown in Figure 2, the aqueous solutions of the PVA hydrogel composites were formed with physical cross-linking points in the form of tiny crystalline regions via intermolecular H-bonds in the process of repeated freezing and thawing. This provided the hydrogel matrix with a three-dimensional network structure with considerable mechanical strength and ability to swell [24,28]. Furthermore, with the BGF evenly dispersed in the gel network, a crystalline region with H-bond interaction between hydroxyl groups of PVA molecules and hydroxyl groups of the GF surface could be observed. 

This hypothesized mechanism was confirmed with the results from FTIR analysis for function group characterization, and the formation of H-bonds between the PVA hydrogel and the phosphate glass fiber was observed; these bonds acted as the physical crosslinking points to further improve the elasticity and strength of the composite hydrogel. Additionally, Chen et al. [24] revealed that the crystallinity of the hydroxyapatite-reinforced PVA hydrogel composite was improved with the adding of hydroxyapatite (HA) particles, due to the intramolecular and intermolecular H-bonds between PVA and HA formed in the process of freezing–thawing, which promoted the crystallization of PVA. A further reasonable explanation could be the nucleating effect of BGF in the hydrogel matrix. Arroyo et al. [29] and Manchado et al. [30] suggested transcrystallinity on the surface of the polymer, bonding to fibers that behaved as effective nucleation agents for the crystallization of the polymer matrix. As such, greater BGF dispersion in the PVA hydrogel matrix could benefit the fiber surface area enabling crosslinking interaction between PVA and BGF, as shown in Figure 2.

### 2.2. Mechanical Performance of Composites during Degradation

The mechanical properties of composites are essential to their application in joint artificial cartilage, especially their initial properties before degradation and consistent capabilities during degradation. With increasing BGF loadings, the composites presented more rapid increasing trends in compressive strength, and ultimately reached higher strength values. As shown in Figure 3B, the compressive strength was significantly increased from 0.22 MPa (PVA) to 0.84 MPa (PBGF-5), 1.52 MPa (PBGF-10) and 3.05 MPa (PBGF-15) when the BGF fiber mass fraction increased from 0 to 15%. Furthermore, the modulus was observed to improve from 0.40 to 2.09 MPa, 2.80, and 3.97 MPa, respectively.

According to the literature, the compressive stress of articular cartilage is about 0.8 MPa [31,32], whilst for high-performance-demand regions, such as the meniscus of the knee, the compressive strength and modulus should be around 1~19 MPa and 3~50 MPa [33,34]. Thus, the composite with a fiber fraction around 15% presented an acceptable performance, matching that of natural articular cartilage. 

During the degradation, the compressive performance of the composites and pure PVA hydrogel were characterized and plotted against degradation time, as shown in Figure 3C,D, respectively. PBGF-15, with the maximum fiber addition, was seen to achieve significantly higher initial compressive strength and modulus when compared to those of the other composites and pure PVA hydrogel. However, the tensile strength of the PBGF-15 composite revealed a consistently steady status within the first 14 days, followed by a rapid decrease by the end of Day 56, falling from 3.05 to 2.08 MPa. For the PBGF-10 and PBGF-15 composites, a steady-state compressive strength by Day 7 was observed, with a further slight decrease by Day 56, and no significant change in the tensile strength was observed for the pure PVA hydrogel during the course of the study. Furthermore, the compressive moduli for the composites were relatively constant up to Day 56, with final values of 0.32, 2.17, 3.10, and 3.49 MPa, for the PVA hydrogel, PBGF-5, PBGF-10, and PBGF-15 composites, respectively.

On the other hand, the scanning electron microscopy (SEM) showed that random chopped fibers were uniformly distributed on the composite surface, resulting in a rough surface with different-sized pores, which could be beneficial to adhesion and growth of cells on the surface of composite. In contrast, a relatively smooth surface without fibers was observed in the PVA hydrogel (Figure 4).

### 2.3. Degradation Behavior of Composites

#### 2.3.1. Mass Variation and Ion Release Behavior 

In order to understand the interaction between biodegradable glass fiber and PVA hydrogel during degradation while immersed in PBS, the mass variation, ion concentration (Ca^2+^, Mg^2+^, BO_3_^3−^ and PO_4_^3^^−^) and pH value of the PBS were characterized and recorded in Figure 5. 

As the swelling behavior of PVA hydrogel corresponds to composite degradation, the mass variation of the pure PVA and composites presented similar variation trends: a rapid increase during the initial 3 days, followed by a considerable decrease by 56 days, whilst the pure biodegradable BGF presented a consistent decrease in mass, reaching a steady status of 78.76% by 56 days (Figure 5A). During the soaking period, the mass variation of the composites was lower than that of the BGF. By the end of the test on Day 56, the residual weight of the PVA was 99.03%, compared to 95.08%, 91.04%, 88.43%, and 78.76%, respectively, for the PBGF-5, PBGF-10, PBGF-15 hydrogel composite, and BGF. When compared with the mass variation of PVA, it was assumed that the degradation of the PVA matrix for the composites was responsible for the mass loss of the composites rather than degradation of BGF. 

Figure 5B showed the pH value of the PBS, which was decreased from 7.50 to 7.22, 6.84, 6.01, and 5.63 for the pure PVA, the PBGF-5, PBGF-10, PBGF-15, and the biodegradable glass fiber (BGF), respectively, by the end of the 56 days. The decrease in pH value for the PVA was attributed to the acidity in the PVA chemical structure, whilst the significant drop in pH value for the BGF in the initial 3 days was caused by the glass degradation, with the release of acidity by the ion species (PO_4_^3−^). Normally, the components in the BGF, such as CaO and MgO, when dissolved into the PBS to release Ca^2+^ and Mg^2+^ ions, could react with the hydroxyl in the water to form Ca(OH)_2_ and Mg(OH)_2_ with alkaline, thus increasing pH, whilst the network components of the glass fiber, including P_2_O_5_ and B_2_O_3,_ could produce PO_4_^3−^ and BO_3_^3−^ ions, generating acidity to reduce the pH value. As such, the effect on pH variation for PBGF composite could be contributed from ions released from PVA hydrogel and BGF. 

When compared with pure BGF, the lesser weight loss and lower pH of the PBS for the PBGF composites indicated that the PVA hydrogel could protect the BGF from PBS degradation and improve the stability of the system. This was also confirmed by ICP analysis of the ions released from the composites into the PBS (Figure 5C–F). The data presented a similar trend, where a greater amount of BGF doped into the PVA hydrogel speeded up the ion release rate, resulting in the variations in weight loss and pH changes. Additionally, the cumulative amounts of PO_4_^3−^, BO_3_^3−^, Ca^2+^, and Mg^2+^ ions in the PBS solution showed the same trend for all groups of specimens. 

As such, the properties of the controllable ion release kinetics of PBGF composites showed the PVA hydrogel had sustained capabilities, proving it more suitable for cartilage repair than previous materials. In the preliminary study, the biodegradable-glass-fiber-reinforced polylactic acid (PLA) composites, with excellent mechanical performance matching the requirements of bone repair, were observed to degrade rapidly when immersed in PBS [35]. Almost all fibers were degraded by the 15th day, mainly due to the acidity ions aggregated in the interface with autocatalytic degradation issues. For further analysis, the construction of BGF in the PVA hydrogel matrix could be effectively protected with better sustainability via ion-exchange adjustment to keep the dynamic balance of pH and ion concentration between interface region and PBS solution (Figure 5G). 

Therefore, the fact that the pure BGF exhibited higher mass loss with a lower pH value than the PBGF composites should be attributed to the PVA-matrix protection, whereas the mass variation in the composites could be mainly attributed to the mass loss of the PVA matrix, with the partial influence of BGF degradation. 

#### 2.3.2. Dynamic Compressive Behavior during Degradation 

The dynamic compressive behaviors of the pure PVA and the PBGF composites were tested using sinusoidal cyclic compression simulating the compression between bone joint and cartilage (shown in Figure 6). The stress retention rate (σ_n_/σ_1_) was defined as the ratio between the maximum stress of each cycle (σ_n_) and the first cycle (σ_1_). 

Obviously, for all samples (PVA, PBGF-5, PBGF-10, and PBGF-15), with increasing cycle times, the stress retention rate σ_n_/σ_1_ for each cycle first decreased rapidly and then gradually reached equilibrium. For the initial period of degradation, the plots of σ_n_/σ_1_ could be separated into three stages: for the initial stage of three cycle times, σ_n_/σ_1_ decreased rapidly, indicating destruction of the delicate structure of the hydrogel. Afterward, the σ_n_/σ_1_ declined slowly during the second stage and finally presented a tiny decrease in the third stage of two final cycles. By the end of the cycle, the loss of σ_n_/σ_1_ was almost 10.55% for the PVA hydrogel, while it declined to 1.03% when the BGF mass fraction was increased to 15% (PBGF-15), indicating an improvement in recoverable ability for the composites. The introduction of BGF into the PVA hydrogel matrix could generate physical crosslinking points in the hydrogel, thus further improving the elasticity and strength of the composite, and helping to improve its anti-fatigue ability when the composite is used in joint motions, such as jumping and running.

During the degradation period over 56 days, increases in the loss of σ_n_/σ_1_ could be observed for all samples. By the end of the degradation period on Day 56, the losses of σ_n_/σ_1_ were 16.23, 15.54, 11.84 and 4.39 for PVA hydrogel, PBGF-5, PBGF-10 and PBGF-15 composites, respectively. This could be attributed to the degradation of the fiber, resulting in loss of interface between the BGF and the PVA hydrogel matrix, with poor load-transfer capability between the matrix and the stiff fibers under compressive load. However, the decrease in the loss of σ_n_/σ_1_ with the addition of fiber could be due to the increase in the surface area for better interface, resulting in more fibers for load transfer. As such, the PBGF-15 composites with the highest fiber mass fraction presented better anti-fatigue capability with stable structure when compared with others.

### 2.4. Cell Study In Vitro

In order to evaluate the feasibility of hydrogel composites reinforced by biodegradable glass fiber as biomimetic implants for cell survival, the chondrocytes were seeded on the surfaces of the composite samples for a 96 h culture and stained by a live/dead cell staining kit. As can be seen from Figure 7A, most of the cells were alive (green) after a 96 h cell culture, although a few dead cells (red) could be observed. A mass of prominent filopodia and unidirectional lamellipodia extension were observed for cells cultured in all groups of PBGF composites, whilst a large cluster of cells with lamellipodia were formed in the PVA hydrogel. For further comparison, the cells cultured on the PBGF-15 composite grew more densely than the other groups, indicating that the increase in biodegradable glass fiber was beneficial for the proliferation of cells, due to the ions released from BGF, such as Ca^2+^, Mg^2+^,and PO_4_^3−^, allowing cells to grow and adhere [36]. 

To analyze the proliferation of chondrocytes on the samples of hydrogel composites, a CCK-8 test was performed, and the results (shown in Figure 7B) demonstrated the obvious difference in cell proliferation among the four groups after 96 h, indicating that hydrogel reinforcement by BGF had a positive effect on cell proliferation with no observable cytotoxicity. This was also reported in the preliminary study, confirming that the cell proliferation could improve with the increase in BGF content [37,38]. This study was the preliminary investigation into the biodegradable-glass-fiber reinforcement of PVA hydrogel composites, and was conducted in order to understand the degradation behavior of glass fiber with the protection of hydrogel matrix and to investigate the improvement of composite structure stability for articular cartilage replacement. 

Based on the results, it is evident that biodegradable glass fiber has the explicit capacity to improve the bioactivity of PVA hydrogel with cell adhesion and proliferation of cells. Furthermore, introduction of BGF into hydrogel matrix produces excellent mechanical properties with improvement in anti-fatigue behavior. However, for further research on the area of cartilage replacement, in vivo investigations should be considered in the future.

## 3. Materials and Methods

### 3.1. Material Preparation

#### 3.1.1. Biodegradable Glass Fiber Manufacture

Biodegradable glass fiber of composition 48P_2_O_5_-12B_2_O_3_-15MgO-14CaO-1Na_2_O-10Fe_2_O_3_ was provided by Sinoma Co., Ltd. (Nanjing, China). The phosphorous pentoxide (P_2_O_5_), boric acid (H_3_BO_3_), calcium hydrogen phosphate dihydroate (CaHPO_4_⦁2H_2_O), magnesium hydrogen phosphate trihydrate (MgHPO_4_⦁2H_3_O), sodium dihydrogen phosphate dihydrate (NaH_2_PO_4_⦁2H_2_O), and iron phosphate tetrahydrate (FePO_4_⦁4H_2_O) were obtained from Sinoagent Co., Ltd. (Shanghai, China). All of the precursors were weighed and placed into platinum crucibles at 1200 °C for full melting, and then drawn in industrial scale via a specific melt–drawing spinning process from 100-tipped busing. 

#### 3.1.2. PVA-BGF Composite (PBGF) Preparation

As can be seen from Figure 8, the weighted polyvinyl alcohol (PVA) from Aladdin Reagent (Shanghai, China) was dissolved in deionized water via magnetic stirring at 90 °C for 30 min until the solution was transparent, then cooled down to room temperature with a solution concentration of 20% (wt). The biodegradable glass fibers (BGF) were chopped into lengths of 2 mm and mixed with PVA solution, then moved into the cylinder mold for casting with two-fold freezing/thawing processing (being frozen at −18 °C for 12 h and then thawed at 4 °C for 4 h), to obtain the PVA-based, fiber-reinforced hydrogel composites. The ratios of BGF to final composites were 5, 10, 15 in weight, and the corresponding obtained composites were named PBGF-5, PBGF-10, and PBGF-15, respectively. 

### 3.2. Characterization 

#### 3.2.1. Swelling Tests

The composites were weighted as $W_{O}$ before being immersed in phosphate-buffered saline (PBS) at 37 °C. With regular intervals, the hydrogel was taken out of the PBS solution, dried superficially with tissue and weighted as $W_{S}$, so the swelling rate (Q) was calculated following Equation (1).
(1)$$Q=\frac{W_{S}-W_{O}}{W_{O}}\;$$

#### 3.2.2. ATR-IR Spectroscopy

The specific chemical groups of the BGF surface, the PVA matrix, and the interface of the PVA-BGF hydrogel were characterized by attenuated total reflectance infrared spectroscopy (ATR-IR) within the range from 4000 to 500 cm^−1^ (Thermo Scientific Nicolet iS20).

#### 3.2.3. Differential Scanning Calorimetry (DSC)

The thermal analysis of the composite hydrogels was conducted using DSC (NETZSCH DSC214), with sample weights of 10 mg and a heating range from room temperature to 250 °C at a heating rate of 20 °C/min under nitrogen flow of 50 mL/min flow rate. Melting temperature (T_m_) and glass transition temperature (T_g_) were determined from the DSC curve, while the percentage of the crystallinity of the PVA hydrogel matrix for each composite was calculated using Equation (2).
(2)$$\%\;of\;Crystallinity=\frac{\Delta H_{m}}{\Delta H_{m100\%}}\times 100\%$$
where $\Delta H_{m}$ was heat of melting of each sample from DSC curve, and $\Delta H_{m100\%}$ was heat of melting of 100% crystalline PVA, 138.6 J/g [39].

#### 3.2.4. Thermogravimetric Analysis (TGA)

The TGA for composites was carried out via the thermal analyzer (NETZSCH TG209) in the temperature ranging from 25 to 800 °C under the heating rate of 10 °C/min; the gas phase was air flow at 50 mL/min.

#### 3.2.5. Mechanical Behavior

The composites were shaped into cylinders (15 mm in diameter × 5 mm) for compressive strength and modulus analysis, with cross-head speed of 1 mm/min. The composite samples were compressed to 70% of their own deformation with preloading of 0.1 N to ensure contact. All mechanical testing were performed by universal testing machine (UTM, UTM4304X) from Suns China.

#### 3.2.6. Dynamic Compressive Behavior

The cyclic compressive behavior of the PBGF composites were measured using a universal testing machine (UTM, UTM4304X) from Suns China. All the samples in the shape of cylinders (15 mm in diameter × 5 mm) were preloaded with 0.1 N to ensure contact. The cyclic compression was performed with an amplitude of 50% strain and rate of 1 mm/min for 10 cycles. 

#### 3.2.7. Degradation Behavior In Vitro

The composite samples, conforming to the dimensions for mechanical analysis, were prepared for degradation analysis. They were placed into phosphate buffered saline (PBS) solution at 37 °C, based on the ratio of 1 g of composite to 30 mL PBS, and time points for checking were 0, 1, 3, 7, 14, 21, 28, and 56. The pH of the PBS solution was measured by pH meter (SX-610), and the weights of the composites for each time point were recorded after drying by tissue. After that, the compressive properties of composite samples were measured in accordance with “3.2.5 Mechanical Behavior”. For further comparison, the pure biodegradable glass fibers were chopped into average lengths of 50 mm and prepared with 100 mg for each vial filled with 30 mL PBS. The weight loss and pH value were measured at the same time points for synchronous comparison. Additionally, the concentrations of P, B, Ca, and Mg releasing ions in the PBS solution for each specimen at each time point were determined by inductively coupled plasma optical emission spectroscopy (ICP-OES; PE Avio 200).

#### 3.2.8. Scanning Electron Microscope (SEM)

The SEM images were taken from cross-section morphology of the composites after tensile performance analysis with Pharos G1 microscope, in order to evaluate the degradation behavior and interface morphology. The freeze-dried samples were loaded onto the surface of the specimen holder with a thin gold film coating by sputtering, and the images were obtained using an accelerating voltage of 10 KV. 

#### 3.2.9. Cell Study In Vitro

The hydrogel composite samples were shaped into cubic dimensions of 5 × 5 × 5, and transferred to 48-well plates for sterilization under ultraviolet light overnight, then incubated in cell culture medium for 2 h before cell seeding. The chondrocyte suspension within 10^4^ cells/well in Dulbecco’s Modified Eagle Medium (DMEM) supplemented with 10% fetal bovine serum (FBS) and 1% penicillin/streptomycin) was seeded onto sample surfaces for specific intervals at 37 °C in 5% CO_2_ incubator. 

After 24 h culture, the original medium in the wells of the cell-culture plate was replaced with 1 mL/well of serum-free medium with 10% CCK-8, then incubated for 3 h at 37 °C. The absorbance was measured using a Microplate Reader (Infinite M200Pro Tecan, Swiss) at 450 nm. Additionally, the composite specimens with cells collected and assessed by live/dead staining were observed by the confocal laser scanning microscope (U-TBI90 Olympus, Japan). Cell/hydrogel constructs, after being cultured for 24 h, were washed with PBS and stained with the mixed solution containing 1 μg/mL fluorescein diacetate (FDA) and 1 μg/mL propidium iodide (PI) for 3 min. After being rinsed with PBS, the constructs were observed with CLSM.

## 4. Conclusions

In summary, novel biodegradable-glass-fiber-reinforced PVA hydrogel composites with enhanced biocompatibility and mechanical properties were successfully synthesized. It was observed that hydrogen bonding occurred amongst the hydroxyl groups of the PVA and the interface between the hydrogel and the BGF, resulting in an improvement in interface performance with better load transfer. Additionally, introducing BGF into the PVA hydrogel could lead to better structure stability with higher crystallinity of hydrogel composites, due to the formation of more crosslinking points in the hydrogel network. As such, the anti-fatigue behavior could be improved with the addition of BGF, and the compressive strength and modulus could satisfy the demands placed on articular cartilage. Furthermore, the results from the in vitro cell test revealed a significant improvement in the proliferation of chondrocytes for the composites fabricated by introducing fiber into PVA hydrogel. Therefore, this research provided a new biomaterial with excellent mechanical performance and bioactivity for efficient cartilage repair.

## Figures and Tables

**Figure 1 ijms-23-08717-f001:**
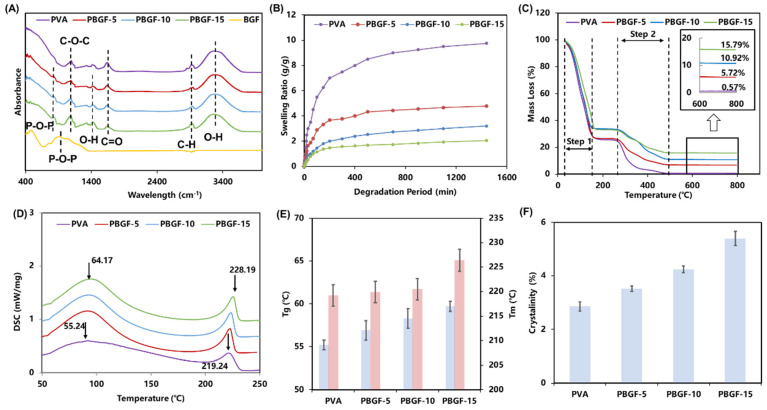
(**A**) ATR-IR spectra of BGF, PVA, and PBGF composites; (**B**) swelling kinetics; (**C**) TGA curve; (**D**) DSC curve; (**E**) glass transition temperature (T_g_) and melting point (T_m_); (**F**) crystallinity percentage for PVA and PBGF composites.

**Figure 2 ijms-23-08717-f002:**
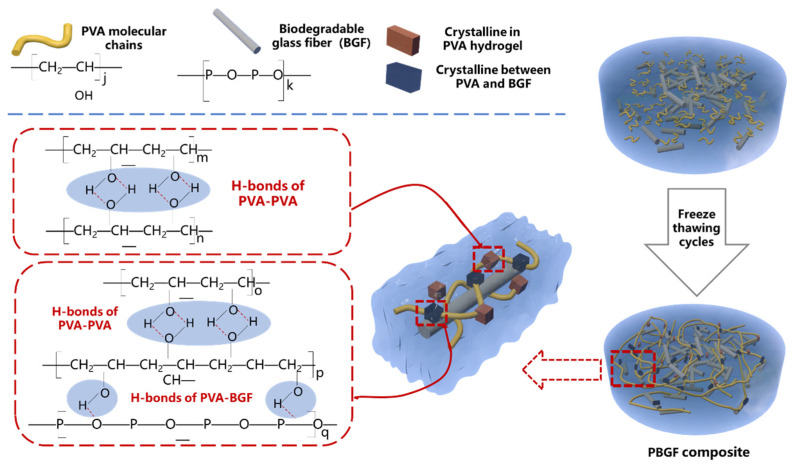
Schematic illustration of microstructure formation for PBGF composites.

**Figure 3 ijms-23-08717-f003:**
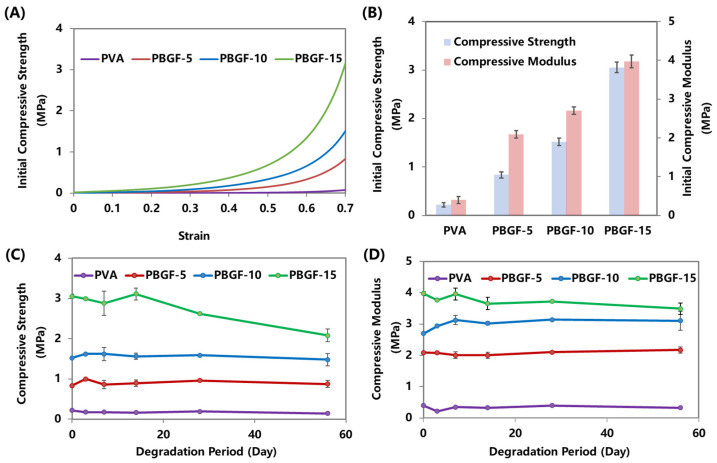
Characterization of mechanical performance before and after degradation for PVA and PBGF composites: (**A**) The compressive strength-strain curves; (**B**) The initial compressive strength and modulus; (**C**,**D**) The compressive strength and modulus during degradation immersed in PBS for 56 days.

**Figure 4 ijms-23-08717-f004:**
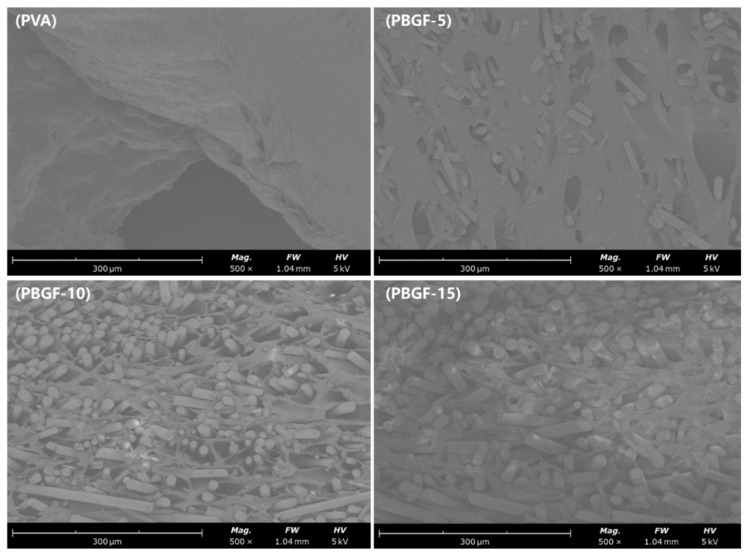
Scanning electron microscopic image of the pure PVA and PBGF composites.

**Figure 5 ijms-23-08717-f005:**
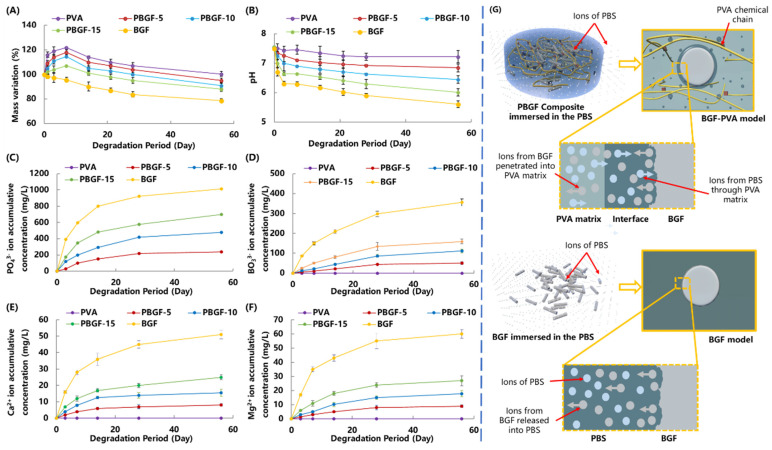
Degradation behavior of composites when immersed in PBS for 56 days: (**A**) The mass variation of PVA hydrogel and PBGF composites; (**B**) The pH value of PBS, where the BGF, PVA and PBGF composites were immersed; The ions of PO_4_^3−^ (**C**), BO_3_^3−^ (**D**), Ca^2+^ (**E**), and Mg^2+^ (**F**) released from BGF, PVA, and composites when immersed in the PBS solution; (**G**) The degradation mechanism diagram with ion movement for composites and pure biodegradable glass fiber when immersed in the PBS solution.

**Figure 6 ijms-23-08717-f006:**
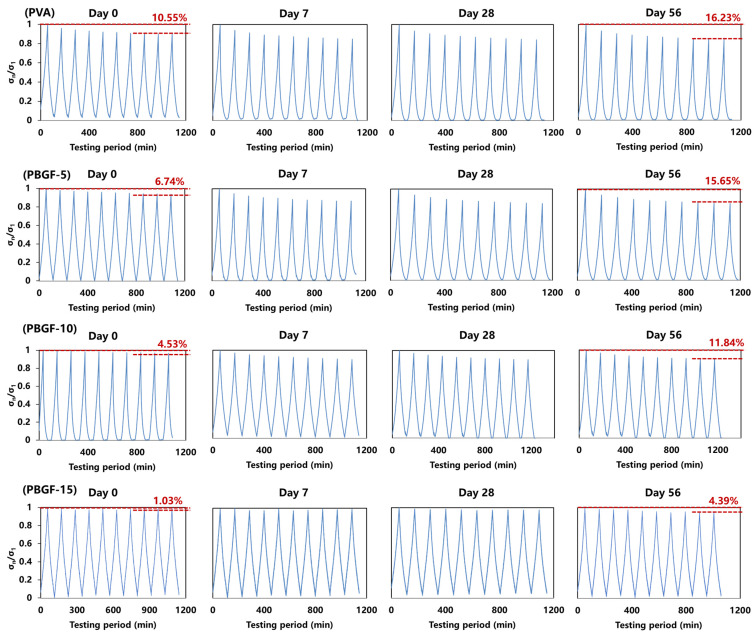
The compressive stress ratio σ_n_/σ_1_ of PVA hydrogel and PBGF composites as a function of time under 10 cycles when immersed in PBS for 56 days.

**Figure 7 ijms-23-08717-f007:**
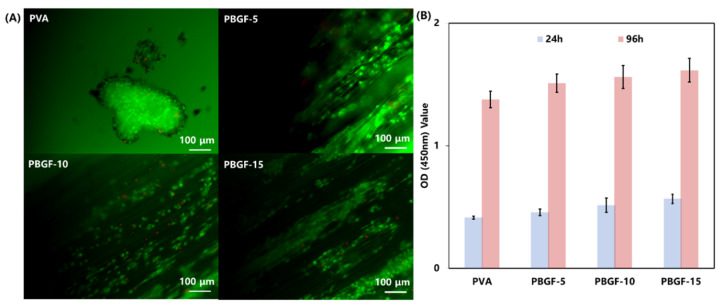
(**A**) Live/dead staining of chondrocytes seeded on hydrogel after being cultured for 96 h. The live cells are stained green by FDA, while the dead cells are stained red by PI. The scale bar is 100 nm. (**B**) CCK-8 test for the proliferation of chondrocytes on the hydrogel and composites by 24 h and 96 h. Data are expressed as means ± SD (*n* = 5).

**Figure 8 ijms-23-08717-f008:**
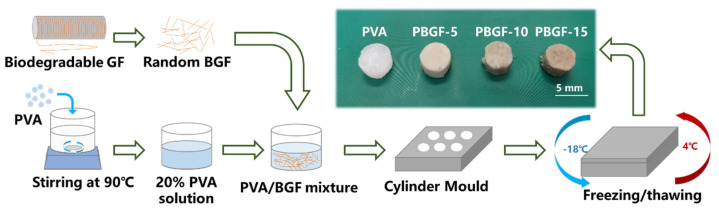
The manufacture process of BGF-reinforced PVA hydrogel composites.

**Table 1 ijms-23-08717-t001:** The FTIR functional groups and corresponding wavenumber.

Functional Group	Wavenumber (cm^−1^)	Note	Ref
P-O-P	750–880	The symmetric mode of P-O-P in biodegradable glass fiber	[23]
C-O-C	1143	The C-O-C stretching vibration of PVA	[19,20]
O-H	1300–1380	The stretching vibration of the H-bond for interface	[24,25]
C=O	1750	The C=O stretching vibration of PVA matrix	[22]
C-H	2920	The C-H stretching vibration of PVA matrix	[21]
O-H	3100–3600	The stretching vibration of the H-bond for PVA matrix	[22]

## Data Availability

Not applicable.

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
