# Peer review of "Biodegradable-Glass-Fiber Reinforced Hydrogel Composite with Enhanced Mechanical Performance and Cell Proliferation for Potential Cartilage Repair"

_ijms, 2022, doi:10.3390/ijms23158717_

Round 1

Reviewer 1 Report

I have read the manuscript “Biodegradable glass fiber reinforced hydrogel composite with 2 enhanced mechanical performance and cell proliferation for 3 potential cartilage repair” by Chenkai Zhu and coworker submitted to International Journal of  Molecular Sciences.

The authors realized composite hydrogel made of poly vinyl alcohol (PVA) reinforced with biodegradable glass fibers (BGF) for potential artificial cartilage repair. They carried out different types of characterization to test the long-term mechanical stability and the cytocompatibility of the composite material.

The paper is interesting and results sound reasonable. Nevertheless, in my opinion, the manuscript needs editing improvements before its publication. In particular:

1.      The formatting of the whole text must be correct. Paragraph and sub-paragraphs titles have different formatting;

2.       Acronyms should be defined the first time they are used (PE at line 57 and HA at line 137) and they should always be written in the same way (PBGF, BGF);

3.      The authors should mention which type of characterization was used in the abstract (line 19);

4.      Please rephrase the sentences at lines 37 and 64;

5.      Please add reference about hydrogel matrix with BGF at line 55;

6.      Even if the composition of the different composition of PBGF samples are reported in the section of material and methods, the authors should specify their composition the first time they are used at line 76;

7.      Please, at lines 78-79 the authors should indicate which value of absorption peak corresponds to each functional group;

8.      Please, at lines 89-91 the authors could avoid to repeat “PVA hydrogel and composites” several times.

9.      There is no reference of the Table 1 in the text (line 101);

10.  There is an error in the numbering of the subsection at line 102;

11.  Please standardize the nomenclature of the Figures 1 and 3 in the text;

12.  The authors could add a discussion of the results of the TGA analysis (lines 109-111);

13.  The authors could discuss the increase of crystallinity for the composite sample (lines 120-121);

14.  In the Figure 3(A) the secondary y axis indicating the modulus has GPa as the unit of measurement, while in the text you write MPa. For a better understanding of the graph you could rotate the title of the secondary y axis and you could indicate the color of the columns corresponding to the Initial Compressive Strenght (blue) and the color of the column indicating the Modulus (pink). Besides, in the text there is no discussion of the Figure 3(B).

15.  There is a lot of difference between the values of modulus at line 155. Maybe it should be between the two values or check that they are correct.

16.  Lines 169-170: maybe the other composites are PBGF-5 and PBGF-10.

17.  In the Figure 5 the authors describe the degradation mechanism and not the mechanical properties (line 186);

18.  The authors should specify the analyzed ions (lines 186-187);

19.  Please correct the numbering of the figures 5;

20.  There are two Figure 6;

21.  There is no reference to Figure 7B and the relative explanation of the graph.

22.  The authors could discuss better the conclusions.

Author Response

The response to Reviewer 1

Dear Reviewer

Thank you so much for your kindly review and important suggestion.

I have modified the manuscript following your guidance and response your comments showing below. 

  1. The formatting of the whole text must be correct. Paragraph and sub-paragraphs titles have different formatting;

Response: The paragraphs have been corrected with uniform format.

  1. Acronyms should be defined the first time they are used (PE at line 57 and HA at line 137) and they should always be written in the same way (PBGF, BGF)

Response: The detail has been corrected with correct acronyms.

  1. The authors should mention which type of characterization was used in the abstract (line 19);

Response: The characterization in abstract has been provided with detail testing.

  1. Please rephrase the sentences at lines 37 and 64;

Response: The sentences have been rephrased.

  1. Please add reference about hydrogel matrix with BGF at line 55;

Response: Actually, we cannot find the research from literature about the BGF reinforced hydrogel, only the study related to the bioactive glass-PVA hydrogel composite. That is why I hope to do this work, understand the degradation behavior and interaction between BGF and hydrogel.

  1. Even if the composition of the different composition of PBGF samples are reported in the section of material and methods, the authors should specify their composition the first time they are used at line 76

Response: The detail of composites has been mentioned in the line 76.

  1. Please, at lines 78-79 the authors should indicate which value of absorption peak corresponds to each functional group;

Response: The functional group corresponding to the weave number has been described in detail and also recorded in Table 1.

  1. Please, at lines 89-91 the authors could avoid to repeat “PVA hydrogel and composites” several times.

Response: The details have been modified.

  1. There is no reference of the Table 1 in the text (line 101);

Response: The reference has been provided.

  1. There is an error in the numbering of the subsection at line 102;

Response: The error was corrected.

  1. Please standardize the nomenclature of the Figures 1 and 3 in the text;

Response: The Figure number has been rechecked with correct nomenclature.

  1. The authors could add a discussion of the results of the TGA analysis (lines 109-111);

Response: The TGA results was further discussed.

  1. The authors could discuss the increase of crystallinity for the composite sample (lines 120-121);

Response: The crystallinity results have been further discussed in lines 120-121, the also completely discussed with composite structure in the following section. 

  1. In the Figure 3(A) the secondary y axis indicating the modulus has GPa as the unit of measurement, while in the text you write MPa. For a better understanding of the paragraph you could rotate the title of the secondary y axis and you could indicate the color of the columns corresponding to the Initial Compressive Strength (blue) and the color of the column indicating the Modulus (pink). Besides, in the text there is no discussion of the Figure 3(B).

Response: The result of Figure 3A has been corrected following the suggestion, and the results in Figure 3B were discussed.

  1. There is a lot of difference between the values of modulus at line 155. Maybe it should be between the two values or check that they are correct.

Response: the unit for modulus should be MPa, the details have been corrected.

  1. Lines 169-170: maybe the other composites are PBGF-5 and PBGF-10.

Response: The other composites are PBGF-5 and PBGF-10, so the details were corrected to avoid misunderstand.

  1. In the Figure 5 the authors describe the degradation mechanism and not the mechanical properties (line 186);

Response: The details have been corrected following the comments.

  1. The authors should specify the analyzed ions (lines 186-187);

Response: There were lots of ions released from composite. However, the important ions corresponding to cell activity and proliferation are Ca2+, Mg2+, BO33- and PO43-.

  1. Please correct the numbering of the figures 5;

Response: the numbering of details were corrected.

  1. There are two Figure 6;

Response: Sorry, one of Figure 6 was cited in the main text, this mistake was modified.

  1. There is no reference to Figure 7B and the relative explanation of the graph.

Response: The Figure 7B was referred in the text with the corresponded detail.

  1. The authors could discuss better the conclusions.

Response: The conclusion was modified with more discussion on results.

Please let me know if there is any problem need to correct. Many thanks!

Reviewer 2 Report

The manuscript entitled ”” by Zhu et al. is interesting, provides new information, is well written and is suitable for publication in IJMS after a minor revision.

Suggestions:

-          P1, L37: Change “and it  resulted in lack of regenerative factors” to “and it results in lack of regenerative factors” Strange to change the present tense to past tense in the middle of the sentence. Keep present tense.

-          P2, L57: Change “are still unknow yet.” to “are still unknow.”

-          P2, L74. Find the correct figure reference instead of “Error! Reference source not found.”

-          Nice Figure 2!

-          P4, L133: Change “characterization. and the formation” to “characterization and the formation”

-          P5, L145: Change title “Mechanical Performance of composites during degradation” to “Mechanical performance of composites during degradation”

-          P5, L148: Change “Figure 3A-B)” to “Figure 3A-B”

-          P9, L242: Remake Figure 6 that all subfigures fit in the figure!

Author Response

The response to Reviewer 2

Dear Reviewer

Thank you so much for your kindly review and important suggestion.

I have modified the manuscript following your guidance and response your comments showing below. 

  1. P1, L37: Change “and it  resulted in lack of regenerative factors” to “and it results in lack of regenerative factors” Strange to change the present tense to past tense in the middle of the sentence. Keep present tense.

Response: Thank you, I have modified the sentence following the reviewer’s suggestion.

  1. P2, L57: Change “are still unknow yet.” to “are still unknow.”

Response: The sentence was corrected.

  1. P2, L74. Find the correct figure reference instead of “Error! Reference source not found.”

Response: The mistake was corrected.

  1. Nice Figure 2!

Response: Thank you!

  1. P4, L133: Change “characterization. and the formation” to “characterization and the formation”

Response: The mistake was corrected.

  1. P5, L145: Change title “Mechanical Performance of composites during degradation” to “Mechanical performance of composites during degradation”

Response: The detail has been corrected.

  1. P5, L148: Change “Figure 3A-B)” to “Figure 3A-B”

Response: The detail has been corrected.

  1. P9, L242: Remake Figure 6 that all subfigures fit in the figure!

Response: The detail of all Figure have been checked and refit with detail in text..

Please let me know if there is any problem need to correct. Many thanks!